# Prolonged mechanical ventilation in patients with severe COVID-19 is associated with serial modified-lung ultrasound scores: A single-centre cohort study

**Hayato Taniguchi**⏺*, **Aimi Ohya, Hidehiro Yamagata, Masayuki Iwashita, Takeru Abe, Ichiro Takeuchi**

Advanced Critical Care and Emergency Centre, Yokohama City University Medical Centre, Yokohama, Japan

\* tanipan@yokohama-cu.ac.jp

## Abstract

Lung ultrasound (LUS), a rapid, bedside, goal-oriented diagnostic test, can be quantitatively assessed, and the scores can be used to evaluate disease progression. However, little data exists on predicting prolonged mechanical ventilation (PMV) and successful extubation using serial LUS scores. We examined the relationship of PMV with successful extubation in patients with severe coronavirus disease (COVID-19) by using two types of serial LUS scores. One LUS score evaluated both the pleura and lung fields, while the other assessed each separately (modified-LUS score). Both LUS scores were determined for 20 consecutive patients with severe COVID-19 at three timepoints: admission (day-1), after 48 h (day-3), and on the seventh follow-up day (day-7). We compared LUS scores with the radiographic assessment of the lung oedema (RALE) scores and laboratory test results, at the three timepoints. The PMV and successful extubation groups showed no significant differences in mortality, but significant differences occurred on day-3 and day-7 both LUS scores, day-7 RALE score, and day-7 PaO2/FiO2 ratio, in the PMV group (p<0.05); and day-3 and day-7 modified-LUS scores, day-7 C-reactive protein levels, and day-7 PaO2/FiO2 ratio, in the successful extubation group (p<0.05). The area under the curves (AUC) of LUS scores on day-3 and day-7, modified-LUS scores on day-3 and day-7,RALE score on day-7, and PaO2/FiO2 ratio on day-7 in the PMV group were 0.98, 0.85, 0.88, 0.98, 0.77, and 0.80, respectively. The AUC of modified-LUS scores on day-3 and day-7, C-reactive protein levels on day-7, and PaO2/FiO2 ratio on day-7 in the successful extubation group were 0.79, 0.90, 0.82, and 0.79, respectively. The modified-LUS score on day 7 was significantly higher than that on day 1 in PMV group (p<0.05). While the LUS score did not exhibit significant differences. The serial modified-LUS score of patients with severe COVID-19 could predict PMV.

## Introduction

Patients with prolonged mechanical ventilation (PMV) have a higher mortality rate and bear higher costs than those who do not require PMV [1]. The average duration of invasive

**Data Availability Statement:** Data cannot be shared publicly because of data contain potentially sensitive information. Data are available from the

Yokohama City University Institutional Ethics Board (contact via rinri@yokohama-cu.ac.jp) for researchers who meet the criteria for access to confidential data.

**Funding:** The author(s) received no specific funding for this work.

**Competing interests:** The authors have declared that no competing interests exist.

mechanical ventilation in patients with coronavirus disease (COVID-19) admitted to the intensive care unit (ICU) was reported to be approximately 8.4 (95% confidence interval [CI] 1.6–13.7) days; however, in some patients, the use is prolonged [2]. The longer the patient is on ventilatory management, the higher the risk of developing ventilator-induced lung injury, and the lung itself is more damaged [3]. Extracorporeal Membrane Oxygenation (ECMO) cannot save irreversibly damaged lungs and is therefore not recommended [4]. In addition, treatment discontinuation may be considered owing to limited resources during a pandemic [5]. Therefore, it is very important to predict whether patients will require PMV or can be extubated if ventilatory management becomes necessary.

It has been reported that patients with COVID-19 who require PMV exhibit fibrosis on computed tomography (CT) [6]. Although CT is useful for assessing lung severity, it requires transportation of critically ill, invasively ventilated patients to the radiology facilities, and this process is challenging. Lung ultrasound (LUS) is a rapid, bedside, goal-oriented, diagnostic test used to answer specific clinical questions, and its findings have been reported to be consistent with CT findings [7, 8]. Moreover, LUS can be quantitatively assessed, and the serial LUS scores can be used to evaluate disease progression [9, 10]. However, there is little knowledge of association of PMV and successful extubation using serial LUS scores.

Therefore, we evaluated whether the serial LUS scores could be associated with PMV and successful extubation in patients with severe COVID-19 who require invasive mechanical ventilation.

## Materials and methods

### Study design and population

This retrospective, single-centre, observational study included consecutive patients from a hospital designated for treating patients with severe COVID-19. The following patients were included: those with a positive nasopharyngeal reverse transcription polymerase chain reaction for severe acute respiratory syndrome coronavirus 2; those aged >18 years; and those who required mechanical ventilation for over 48 h. The exclusion criteria were as follows: acute heart failure, interstitial pneumonia, other pulmonary diseases affecting image acquisition or suboptimal ultrasound window, missing ultrasound data, and patients' refusal to consent. The study was approved by the institutional ethics board (No. B200200049). The need for written informed consent was waived, as ultrasound scanning of the lungs is considered a routine procedure.

### Patient management

During invasive mechanical ventilation, sedation analgesia was managed at a Richmond Agitation Sedation Score <-3 in patients with a strong respiratory effort, and muscle relaxant was administered if necessary. The patients' respiratory effort was assessed based on the airway occlusion pressure (P0.1) and physical examination, and a P0.1 >4 was considered a strong respiratory effort. If the respiratory effort was calm, for example, a P0.1≤4 and not using accessory respiratory muscles, daily spontaneous awaking trial (SAT) was performed, and the patient managed according to the Pain, Agitation/Sedation, Delirium, Immobility, and Sleep Disruption guidelines [11].

Ventilation management was performed with pressure-controlled ventilation driving pressure <14 and positive end-expiratory pressure (PEEP), based on a high PEEP table, from the acute respiratory distress syndrome (ARDS) net of respiratory frequency <15 [10]. $FiO_2$ was adjusted to $SpO_2$ >93%. The introduction criteria for prone ventilation according to the PROSEVA study were as follows: $FiO_2$ >60% and $PaO_2$/ $FiO_2$ (P/F) ratio <150 [12]. Prone

ventilation was performed for $\geq$16 h [13]. The introduction criteria for veno-venous ECMO according to the EOLIA trial were as follows: FiO2 >80% and P/F ratio <80 for $\geq$6 h [14].

A spontaneous breathing trial was performed after SAT to evaluate extubation, and if the Rapid Shallow Breath Index was <100, the patient was extubated [15]. The final decision to extubate was made by a team including the physician in charge.

## Clinical data and outcomes

Data on patients' demographic characteristics, imaging and laboratory findings, comorbidities, complications, treatment for COVID-19, and outcomes were extracted from electronic medical records. Laboratory tests and chest X-ray results were recorded every day after admission to assess COVID-19 progression and fibrosis. LUS was performed on admission (day 1), third (day 3), and seventh day (day 7). CT was performed at admission and whenever the physician in charge deemed it necessary. The study's primary endpoint was PMV, and the secondary endpoint was successful extubation. PMV was defined as the requirement of mechanical ventilation for >21 days according to the National Association for Medical Direction of Respiratory Care Consensus Conference [1]. Successful extubation was defined as not requiring reintubation for >3 days [16].

## Performing LUS and chest X-ray scoring

LUS examinations were performed using an ultrasound equipment (GE Venue Go) with a 5-12-MHz linear transducer. LUS was performed at six points per hemithorax (superior and inferior regions anteriorly, laterally, and posteriorly), and bilaterally; a total of 12 regions were assessed with the probe placed in the intercostal space to obtain images widely. In each region, LUS signs, including B-lines/consolidation and pleural line abnormalities, were assessed, and the worst LUS signs were scored as each LUS scores.

In this study we used two types of LUS scores. One was the popular LUS score: score 0: A-lines or two or fewer well-spaced B-lines; score 1, three or more well-spaced B-lines; score 2, coalescent B-lines; score 3, tissue-like pattern, which were used to predict ARDS severity, progression, and lung reaeration in previous studies [16]. The sum of the scores in all 12 zones yielded a final score (ranging from 0 to 36). The other scoring system was modified-LUS (m-LUS) score, in which B-lines/consolidations were quantitatively scored as follows: score 0, well-spaced B-lines <3; score 1, well-spaced B-lines $\geq$3; score 2, multiple coalescent B-lines; and score 3, lung consolidation. The pleural line was quantitatively scored as follows: score 0, normal; score 1, irregular pleural line; and score 2, blurred pleural line, which were associated with COIVD-19 severity at admission [17]. The sum of both scores in all 12 zones yielded a final score with a range between 0 and 60.

Both LUS scores were evaluated by two emergency physicians (A.O and H.Y) blinded to the clinical data. They were well-trained in evaluating both LUS scores and were experienced in performing LUS for over 25 cases [18]. Both scores were evaluated independently, after which the final decision was reached by consensus.

The radiographic assessment of the lung oedema (RALE) score was used to evaluate the chest X-ray [19]. To determine the RALE score, each radiograph was divided into quadrants, defined vertically by the vertebral column and horizontally by the first branch of the left main bronchus. Each quadrant was assigned a consolidation score of 0–4, to quantify the extent of the alveolar opacities based on the percentage of the quadrant with the opacification, and a density score of 1–3, to quantify the overall density of the alveolar opacities, unless the consolidation score for that quadrant was 0. The density score (1 = hazy, 2 = moderate, and 3 = dense) allowed for a more quantitative assessment of the density of opacification by

quadrant. To calculate the final RALE score, the product of the consolidation and density score for each quadrant were summed for the final RALE score, ranging from 0 (no infiltrates) to 48 (dense consolidation in >75% of each quadrant). The RALE score was also evaluated by two experienced emergency physicians (A.O and H.Y) blinded to the clinical data. Scores were evaluated independently, after which the final decisions were reached by consensus.

## Statistical analysis

Continuous variables are expressed as the mean ± standard deviation (SD) or median (inter-quartile range), as appropriate. Categorical variables are presented as frequencies (percent-ages). Analysis of variance (ANOVA) of continuous variables were evaluated using the Kruskal–Wallis test or two-way ANOVA owing to non-normally distributed data. Categorical variables were compared using the chi-square test or Fisher's exact test. To estimate outcome predictors, all potential predictors were included in univariate analyses (Mann-Whitney U test). Variables with P <0.05 in the univariate analysis were used in the receiver operator curve (ROC) analysis. The ROC analysis was performed to examine the sensitivity and specificity of prognosis parameters of the outcomes and determine the area under the curve (AUC) with the 95% CI. All statistical analyses were performed using JMP®, Version 15. (SAS Institute Inc., Cary, NC.).

## Results

### Clinical characteristics

We treated 26 patients with severe COVID-19 who required invasive mechanical ventilation, 6 of whom were excluded owing to missing ultrasound data. Twenty patients who met the inclusion criteria were identified, of which, 11 were PMV cases and 8 were successful extubation cases (Fig 1).

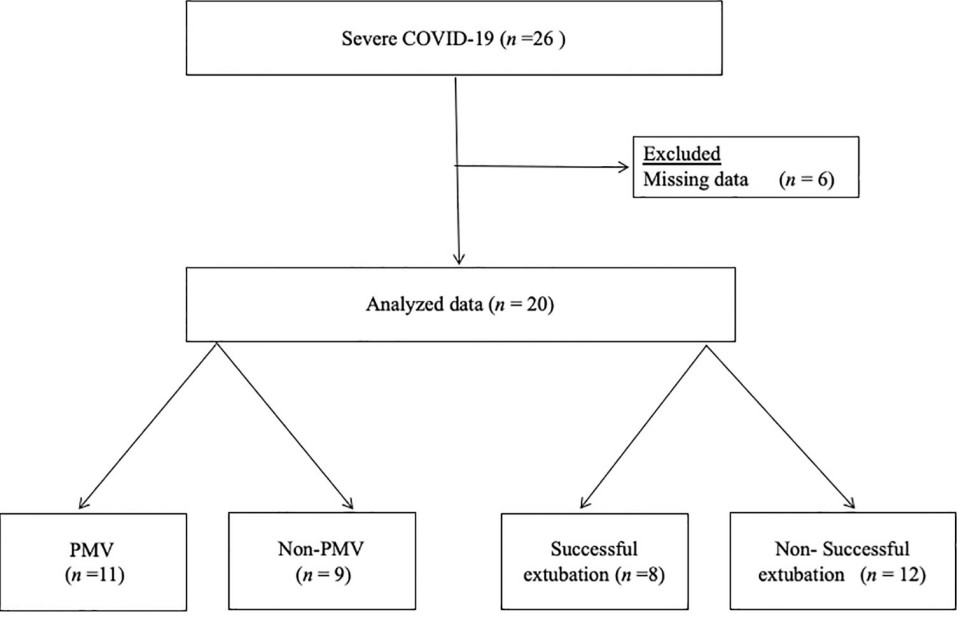

**Fig 1. Patients' selection flow.**

Table 1 summarizes the baseline clinical characteristics and findings of patients with or without PMV. There were no significant differences in age, sex, body mass index, time from first symptom, pre-existing condition, severity score, treatment, complications, outcome, and fluid balance. In imaging and laboratory findings, there were significant differences in the LUS scores on days 3 and 7, m-LUS scores on days 3 and 7, RALE score on day 7, and P/F ratio on day 7 ($p<0.05$). There was no difference with fibrosis on CT at admission.

Table 2 summarizes the baseline clinical characteristics and findings of the patients based on whether they had successful extubation or not. There were no significant differences in clinical characteristics between the groups. In imaging and laboratory findings, there were significant differences in the m-LUS score on day 3 and day 7, CRP levels on day 7, and P/F ratio on day 7 ($p<0.05$).

## The receiver operator curve analysis for predicting outcomes

Variables with $P <0.05$ in the univariate analysis were used in the receiver operator curve (ROC) analysis. Our analyses of the predictors for PMV revealed that the AUCs of the LUS score on day 3, LUS score on day 7, m-LUS score on day 3, m-LUS score on day 7, RALE score on day 7, and P/F ratio on day 7, were 0.88 (95% CI: 0.66–0.96); 0.98 (95% CI: 0.83–1.0); 0.88 (95% CI: 0.66–0.96); 0.98 (95% CI: 0.83–1.0); 0.77 (95% CI: 0.50–0.91); and 0.80 (95% CI: 0.54–0.93), respectively, for PMV ($p<0.05$). The comparison of each AUC was not significantly different (Fig 2a).

For successful extubation, the AUCs of the m-LUS score on day 3, m-LUS score on day 7, CRP levels on day 7, and P/F ratio on day 7, were 0.79 (95% CI: 0.53–0.93); 0.90 (95% CI: 0.59–0.98); 0.82 (95% CI: 0.53–0.95); and 0.79 (95% CI: 0.52–0.93), respectively ($p<0.05$) The comparison of each AUC was not significant different (Fig 2b).

## Analysis of the two types serial LUS scores for each outcome

Fig 3a shows the serial LUS and m-LUS scores on days 1, 3, and 7 with the PMV groups. The m-LUS score on day 7 was significantly higher than that on day 1 ($p<0.05$). While the LUS score did not exhibit significant differences. Fig 3b shows the serial LUS and m-LUS scores on days 1, 3, and 7 with the successful extubation groups. The m-LUS score on day 7 was significantly lower than that of day 1. Though LUS score on day3 and 7 was significantly lower than that of day 1, LUS scores itself did not exhibit significant differences with or without successful extubation.

We showed examples of cases that were monitored using ultrasound and CT (Fig 4).

## Discussion

Several studies on the use of LUS in patients with COVID-19 that use CT as the reference standard have indicated that LUS on admission may predict mortality or the need for invasive mechanical ventilation [20]. However, few studies have assessed whether the serial LUS scores could predict the prognosis of lung injury [9, 10]. This study showed that a higher m-LUS score on day 7 was a predictor for PMV, while a lower m-LUS score on day 7 was a predictor for successful extubation in patients with severe COVID-19.

If patients require PMV, they are usually excluded as candidates for ECMO, and with limited resources during a pandemic, this may be considered a withdrawal of treatment [5]. Therefore, it is very important to predict whether patients will require PMV or can be extubated if ventilatory management becomes necessary. If we could predict the need for PMV early, we could consider transferring the patient to an ECMO centre before ECMO is no

**Table 1. The baseline clinical characteristics and findings of patients with or without PMV.**

| Variable [frequency (%)/median (IQR)] | PMV (n = 11) | Non-PMV (n = 9) | p-value |
|---|---|---|---|
| Age (years) | 66 [56–74] | 65 [55.5–71] | 0.595 |
| Male sex | 10 [50] | 6 [30] | 0.285 |
| BMI | 28.4 [24.9–31.8] | 25 [22.7–28.3] | 0.183 |
| Time from symptom | 9 [5.5–10] | 6 [3–7] | 0.098 |
| *Pre-existing condition* | | | |
| Hypertension | 5 [25] | 7 [35] | 0.196 |
| Diabetes | 6 [30] | 4 [20] | 0.650 |
| Chronic cardiac failure | 1 [5] | 1 [5] | 0.889 |
| Renal insufficiency | 0 [0] | 0 [0] | - |
| Malignancy | 0 [0] | 1 [5] | 0.257 |
| APACHE II score at the time of ICU admission | 20 [16–22] | 19 [15–24] | 0.760 |
| *Treatment* | | | |
| Antiviral therapy, n (%) | 9 [45] | 9 [45] | 0.177 |
| Glucocorticoid therapy, n (%) | 9 [45] | 9 [45] | 0.177 |
| Prone ventilation, n (%) | 6 [30] | 4 [20] | 0.653 |
| Extra-corporeal membrane oxygenation, n (%) | 4 [20] | 1 [5] | 0.195 |
| *Complications* | | | |
| Ventilator-associated pneumonia, n (%) | 4 [20] | 0 [0] | 0.094 |
| Acute kidney injury, n (%) | 1 [5] | 0 [0] | 0.861 |
| *Outcomes* | | | |
| Discharge, n (%) | 10 [50] | 9 [45] | 0.353 |
| Death, n (%) | 1 [5] | 0 [0] | 0.353 |
| *Fluid balance* | | | |
| Total fluid balance on day 3 (mL) | 2734 [1414–4024] | 2672 [2125–3861] | 0.621 |
| day 7 (ml) | 3934 [2205–4518] | 1949 [371–3543] | 0.063 |
| *Imaging findings* | | | |
| Fibrosis on CT at admission, n (%) | 1 [5] | 0 [0] | 0.354 |
| Echocardiography (EF) on admission (%) | 60 [50.0–63.1] | 60 [55.8–66.8] | 0.717 |
| Echocardiography (E/e) on admission | 12.4 [9.86–14.2] | 8.75 [6.35–13.3] | 0.387 |
| LUS score on day 1 | 23 [21–25] | 22 [19–24] | 0.415 |
| day 3 | 24 [22–26] | 19 [17–22] | 0.003 |
| day 7 | 24 [21–26] | 19 [16–22] | 0.01 |
| m-LUS score on day 1 | 36 [35–36] | 35 [31–36] | 0.101 |
| day 3 | 36 [36–39] | 32 [25.5–36] | 0.004 |
| day 7 | 38 [36–41] | 27 [23.5–30] | <0.001 |
| RALE score on day 1 | 20 [19–30] | 22 [19–23] | 0.842 |
| day 3 | 20 [16–30] | 18 [9–24] | 0.252 |
| day 7 | 20 [16–30] | 16 [9.5–17] | 0.045 |
| *Laboratory findings.* | | | |
| Lymphocytes on day 1 (%) | 11 [6.95–13.3] | 8.1 [4.8–12.3] | 0.452 |
| day 3 (%) | 4.5 [3.2–6.95] | 4.4 [3–8.7] | 0.873 |
| day 7 (%) | 5.9 [4.35–10.6] | 4.95 [3.68–13.3] | 0.592 |
| CRP on day 1 (mg/L) | 13.4 [7.04–16.5] | 16.1 [12.7–22.3] | 0.239 |
| day 3 (mg/L) | 6.98 [5.27–15.8] | 7.4 [3.93–10.3] | 0.500 |
| day 7 (mg/L) | 5.87 [4.48–9.28] | 3.83 [2.66–7.10] | 0.102 |
| D-dimer on day 1 (μg/L) | 3.65 [1.58–38] | 1.8 [1.4–2.7] | 0.135 |
| day 3 (μg/L) | 3.6 [2.15–4.4] | 3.2 [2.6–5.4] | 0.682 |

*(Continued)*

**Table 1.**  (Continued)

| Variable [frequency (%)/median (IQR)] | PMV (n = 11) | Non-PMV (n = 9) | p-value |
|---|---|---|---|
| day 7 (μg/L) | 4.1 [2.3–7.5] | 2.4 [1.9–5.5] | 0.482 |
| KL-6 on day 1 (U/mL) | 405 [287–765] | 278 [214–340] | 0.094 |
| day 3 (U/mL) | 753 [398–1049] | 472 [352–1321] | 0.558 |
| day 7 (U/mL) | 787 [444–842] | 509 [365–1016] | 0.366 |
| P/F ratio on day 1 | 220 [108–233] | 144 [116–225] | 0.381 |
| day 3 | 220 [161–266] | 250 [181–298] | 0.381 |
| day 7 | 225 [170–266] | 300 [245–378] | 0.023 |

PMV, prolonged mechanical ventilation; IQR, interquartile range; BMI, body mass index; APACHE II, Acute Physiologic Assessment and Chronic Health Evaluation II; ICU, intensive care unit; CT, computed tomography; EF, ejection fraction; E/e', the ratio between early mitral inflow velocity and mitral annular early diastolic velocity; LUS, lung ultrasound; m-LUS, modified-lung ultrasound; RALE, The radiographic assessment of the lung oedema; P/F ratio, PaO2/FiO2 ratio; CRP, C-reactive protein.

longer applicable. Furthermore, the ability to predict PMV allows for the appropriate allocation of medical resources, including ICU beds.

Gattinoni et al. reported variations in the respiratory mechanic profiles of invasively ventilated patients with COVID-19 pneumonitis, and the following two clinical phenotypes were identified: (1) type L and (2) type H, [21]. The transition from type L to type H may be because of worsening of COVID-19 severity, or an injury caused by high-stress ventilation and patient self-inflicted ventilation (P-SILI) [22]. The depth of the negative intrathoracic pressure may also play a key role in the phenotype shift. If P-SILI is a concern in patients with COVID-19, early intubation is recommended, and adequate sedation and analgesia should be administered to suppress spontaneous breathing [23]. However, excessive sedation and analgesia may result in unsuccessful extubation, which is a risk factor for PMV [24]. We have to evaluate how long the lungs should be rested and when the lungs should be used every day.

Follow-up CT in ARDS patients, including patients with COVID-19, could demonstrate the progression of lung pathology. Pulmonary fibroproliferation, assessed using CT, in patients with ARDS, which is induced by COVID-19 induces, predicts increased mortality and increased susceptibility to multiple organ failure, including ventilator dependency and its associated outcomes [25]. However, in a pandemic, the transportation of critically ill ventilated patients to radiology facilities is challenging, especially for ECMO-managed patients [7, 8, 10]. LUS is a fast, non-invasive, sensitive, and quantitative tool to assess multiple pulmonary pathologies, such as pulmonary oedema, pneumonia, and interstitial lung disease [26]. Furthermore, de Almeida Monteiro et al. showed a histological background that supports the fact that LUS can be used to characterize the progression and severity of lung damage in severe COVID-19 [27]. Therefore, LUS may have very useful imaging findings in patients with COVID-19, which are consistent with CT and pathologic findings.

This study showed no difference in water balance or cardiac function according to the outcome. Therefore, we believe that the worsening of serial LUS scores can be used to evaluate lung injury, such as fibrosis, and not wet lung. Moreover, we analysed two types of serial LUS scores and showed that m-LUS score is more sensitive than LUS score for assessing outcomes. We thought that m-LUS score, which assesses pleural line changes in detail, has more association with this study outcome. Regarding ventilator-associated pneumonia (VAP), only four patients were observed in the PMV group, while there was none in the non-PMV group, but this was not statistically different. In addition, the onset of VAP occurred after 1 week of admission, when our scorings were not performed. While sialylated carbohydrate antigen KL-

**Table 2. The baseline clinical characteristics and findings of patients with or without successful extubation.**

| Variable [frequency (%)/median (IQR)] | Successful extubation (n = 8) | Non- Successful extubation (n = 12) | p-value |
|---|---|---|---|
| Age (years) | 64.5 [54.3–68.5] | 69 [56.3–74] | 0.376 |
| Male sex | 5 [25] | 11 [55] | 0.255 |
| BMI | 25.3 [24.3–28.3] | 26.9 [24–31.4] | 0.537 |
| Time from symptom | 9 [4.8–10] | 6 [3.3–9.3] | 0.349 |
| *Pre-existing condition* | | | |
| Hypertension | 6 [30] | 6 [30] | 0.373 |
| Diabetes | 3 [15] | 7 [35] | 0.649 |
| Chronic cardiac failure | [5] | 1 [5] | 0.889 |
| Renal insufficiency | 0 [0] | 0 [0] | - |
| Malignancy | 0 [0] | 1 [5] | 0.402 |
| APACHE II score at the time of ICU admission | 17.5 [15–23] | 20 [17–23] | 0.756 |
| *Treatment* | | | |
| Antiviral therapy, n (%) | 8 [40] | 10 [50] | 0.497 |
| Glucocorticoid therapy, n (%) | 8 [40] | 10 [50] | 0.497 |
| Prone ventilation, n (%) | 3 [15] | 7 [35] | 0.649 |
| Extra-corporeal membrane oxygenation, n (%) | 1 [5] | 4 [20] | 0.603 |
| *Complications* | | | |
| Ventilator-associated pneumonia, n (%) | 0 [0] | 4 [20] | 0.117 |
| Acute kidney injury, n (%) | 1 [5] | 0 [0] | 0.861 |
| *Outcomes* | | | |
| Discharge, n (%) | 8 [40] | 11 [55] | 0.304 |
| Death, n (%) | 0 [0] | 1 [5] | 0.304 |
| *Fluid balance* | | | |
| Total fluid balance on day 3 (mL) | 2933 [2016–3136] | 2754 [1474–4148] | 0.938 |
| day 7 (ml) | 2016 [370–3682] | 2130 [1474–4485] | 0.063 |
| *Imaging findings* | | | |
| Fibrosis on CT at admission, n (%) | 0 [0] | 1 [5] | 0.402 |
| Echocardiography (EF) on admission (%) | 61.4 [50–66.7] | 54.4 [50–63.1] | 0.713 |
| Echocardiography (E/e) on admission | 9.48 [6.35–13.3] | 12.2 [9.86–14.2] | 0.385 |
| LUS score on day 1 | 23 [21–26] | 22 [20–24] | 0.461 |
| day 3 | 20 [18–24] | 23 [21–25] | 0.245 |
| day 7 | 20 [17–23] | 23 [20–25] | 0.3 |
| m-LUS score on day 1 | 34 [32–36] | 34 [35–36] | 0.371 |
| day 3 | 32 [30–36] | 36 [3–39] | 0.027 |
| day 7 | 28 [26–31] | 37 [36–41] | 0.003 |
| RALE score on day 1 | 22 [22–24] | 22 [17–29] | 0.640 |
| day 3 | 18 [10–24] | 21 [15–27] | 0.614 |
| day 7 | 14 [10–18] | 21 [14–30] | 0.147 |
| *Laboratory findings.* | | | |
| Lymphocytes on day 1 (%) | 7.44 [4.7–11.9] | 11.3 [8.48–13.3] | 0.328 |
| day 3 (%) | 3.75 [2.9–6.88] | 4.75 [3.2–8.4] | 0.447 |
| day 7 (%) | 4.3 [3.55–12.8] | 7.3 [4.63–11.5] | 0.297 |
| CRP on day 1 (mg/L) | 15.9 [12.7–21.1] | 13.8 [7.81–21.2] | 0.488 |
| day 3 (mg/L) | 6.68 [3.76–9.14] | 7.25 [5.47–23.2] | 0.247 |
| day 7 (mg/L) | 3.83 [2.64–4.79] | 6.54 [4.48–12.3] | 0.017 |
| D-dimer on day 1 (μg/L) | 3.65 [1.58–38] | 1.8 [1.4–2.7] | 0.135 |
| day 3 (μg/L) | 7.44 [4.7–11.9] | 3.2 [2.6–5.4] | 0.682 |

*(Continued)*

**Table 2.** (Continued)

| Variable [frequency (%)/median (IQR)] | Successful extubation (n = 8) | Non- Successful extubation (n = 12) | p-value |
|---|---|---|---|
| day 7 (μg/L) | 3.75 [2.9–6.88] | 2.4 [1.9–5.5] | 0.482 |
| KL-6 on day 1 (U/mL) | 282 [212–560] | 382 [280–750] | 0.160 |
| day 3 (U/mL) | 592 [419–1625] | 692 [355–1029] | 0.758 |
| day 7 (U/mL) | 545 [394–1267] | 705 [397–835] | 0.673 |
| P/F ratio on day 1 | 148 [125–228] | 210 [97.8–232] | 0.754 |
| day 3 | 230 [15–295] | 225 [163–287] | 0.643 |
| day 7 | 302 [242–379] | 227 [177–270] | 0.034 |

PMV, prolonged mechanical ventilation; IQR, interquartile range; BMI, body mass index; APACHE II, Acute Physiologic Assessment and Chronic Health Evaluation II; ICU, intensive care unit; CT, computed tomography; EF, ejection fraction; E/e', the ratio between early mitral inflow velocity and mitral annular early diastolic velocity; LUS, lung ultrasound; m-LUS, modified-lung ultrasound; RALE, The radiographic assessment of the lung oedema; P/F ratio, PaO2/FiO2 ratio; CRP, C-reactive protein.

6 (KL-6) is usually used as a biomarker to evaluate lung fibrosis and can predict severity in patients with COVID-19, there was no significant difference between PMV and non-PMV in our study [28]. The possible reason for this as compared with a previous report, may be that our study included only severely ill patients.

There are few reports of patients with COVID-19 who met the usual extubation criteria but were subsequently reintubated [9, 29]. Moreover, CT at the time of reintubation showed progressive lung fibrosis [29]. In our study, three patients in the PMV group met our extubation criteria and were once extubated but reintubated within three days. The reason for the

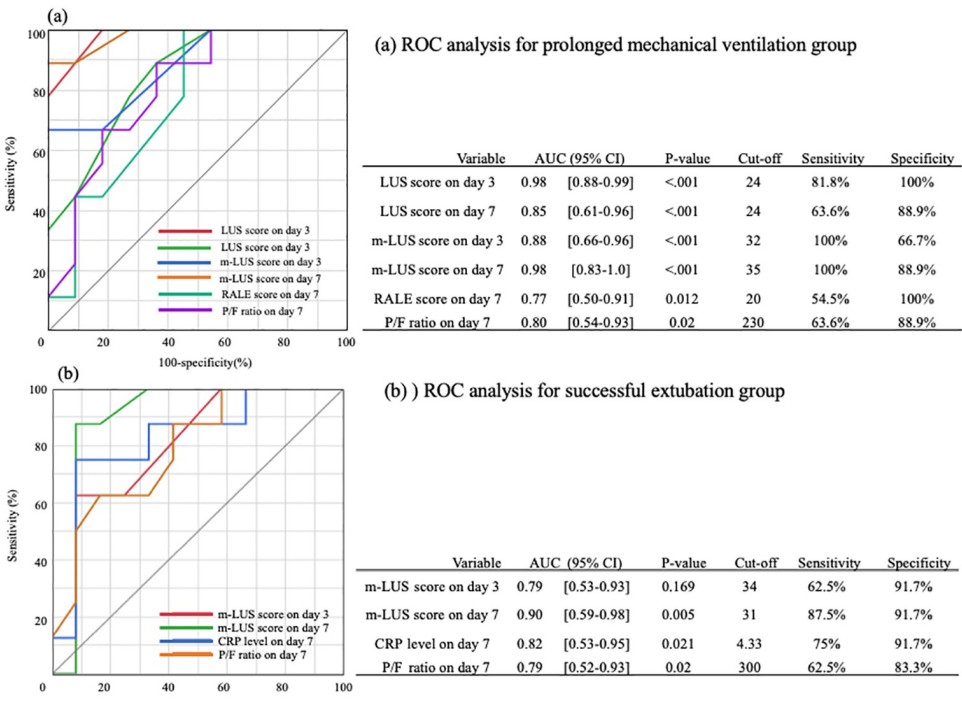

**Fig 2. The receiver operating characteristic curve analysis for predicting outcomes.** AUC, area under the curve; CI, confidence interval; LUS, lung ultrasound; RALE, radiographic assessment of the lung edema; P/F, PaO2/ FiO2, CRP, C-reactive protein.

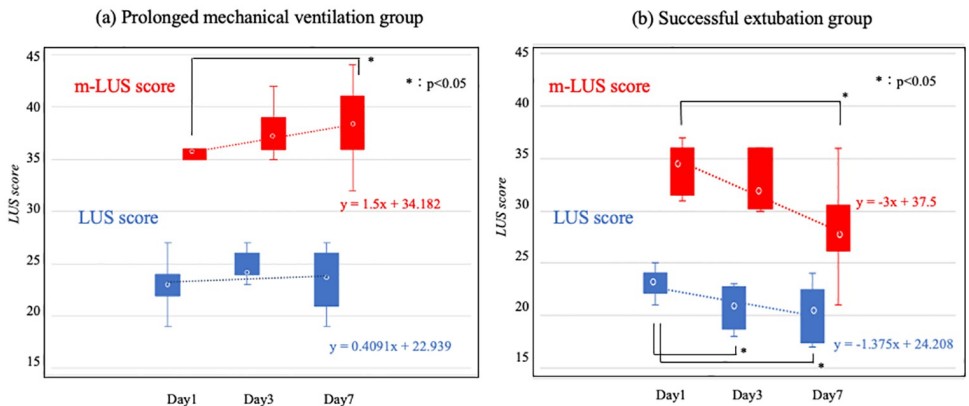

**Fig 3. Analysis of the two types serial LUS scores for each outcome.** LUS, lung ultrasound; m-LUS, modified-lung ultrasound; PMV, prolonged mechanical ventilation.

reintubation could be an exacerbation of the respiratory workload owing to the lung fibrosis. Recently, it has been reported that the success rate of extubation is higher when respiratory effort and diaphragmatic muscle strength are added to the evaluation, besides conventional extubation criteria [30]. Based on our results and previous reports of ultrasound evaluation of the diaphragm, we believe that ultrasound assessment may be considered in future extubation criteria.

This study had some limitations. This was a single-center study with a relatively limited sample size; this could limit the generalizability of our results. Secondly, it is suggested that

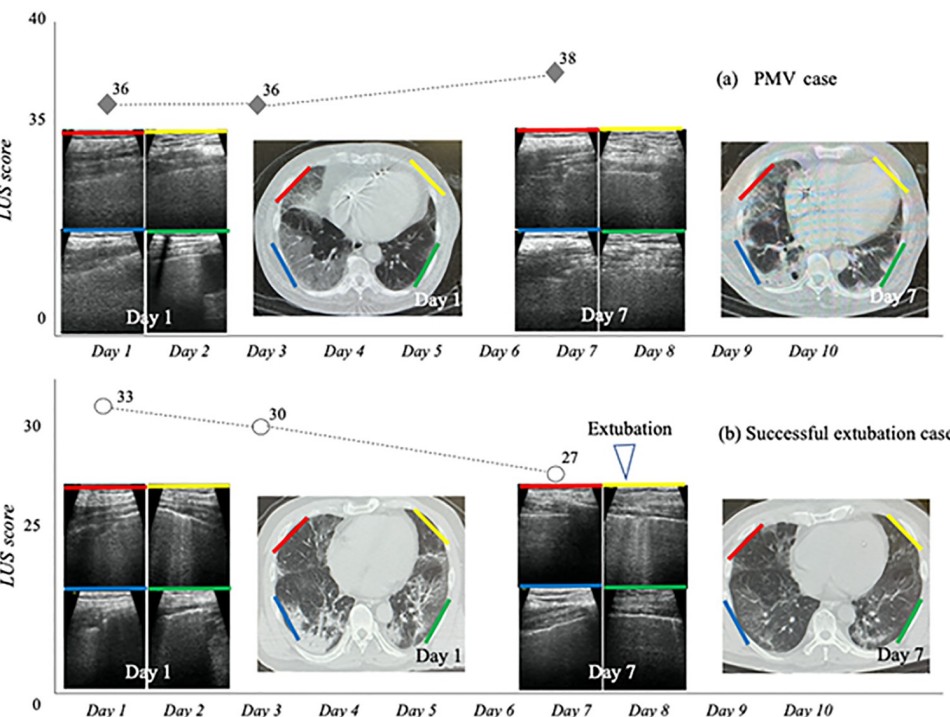

**Fig 4. Examples of cases with PMV and successful extubation that were monitored using ultrasound and CT scans.** m-LUS, modified-lung ultrasound; PMV, prolonged mechanical ventilation.

respiratory muscle strength, including diaphragmatic functions, affect PMV and successful extubation; however, this was not assessed in this study. Finally, we did not perform chest CT at day3 and 7 in all cases. Thus, to evaluate the progression of lung pathology, such as lung fibrosis, the daily comparisons between the LUS score and chest CT could not be performed in this study. Therefore, a longitudinal study that would continuously measure LUS scores and CT with progression of lung fibrosis is required in future studies.

## Conclusions

Patients with PMV have a higher mortality rate and bear higher costs. It is very important to predict whether patients will require PMV with limited medical resources due to the COVID-19 pandemic. This study showed that a higher m-LUS score on day 7 was a predictor for PMV, while a lower m-LUS score on day 7 was a predictor for successful extubation in patients with severe COVID-19.

## Acknowledgments

We thank our colleagues in the Advanced Critical Care and Emergency Centre, Yokohama City University Medical Centre. This manuscript was previously posted on a scholarly collaboration network as ResearchGate, entitled 'Do Serial Lung Ultrasound Scores Predict Prolonged Mechanical Ventilation in Patients With Severe COVID-19? A Single-center Retrospective Cohort Study' [29].

## Author Contributions

**Conceptualization:** Hayato Taniguchi, Aimi Ohya, Hidehiro Yamagata, Masayuki Iwashita, Ichiro Takeuchi.

**Data curation:** Hayato Taniguchi, Aimi Ohya, Hidehiro Yamagata.

**Formal analysis:** Hayato Taniguchi, Aimi Ohya, Hidehiro Yamagata, Takeru Abe.

**Methodology:** Hayato Taniguchi, Takeru Abe.

**Supervision:** Masayuki Iwashita, Takeru Abe.

**Validation:** Masayuki Iwashita, Ichiro Takeuchi.

**Writing – original draft:** Hayato Taniguchi.

**Writing – review & editing:** Hayato Taniguchi.

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
