## [Decision Letter · Decision Letter 0]

4 May 2022

PONE-D-22-08025Prolonged mechanical ventilation in patients with severe COVID-19 is associated with serial lung ultrasound scores: A single-centre cohort studyPLOS ONE

Dear Dr. Taniguchi,

Thank you for submitting your manuscript to PLOS ONE. After careful consideration, we feel that it has merit but does not fully meet PLOS ONE’s publication criteria as it currently stands. Therefore, we invite you to submit a revised version of the manuscript that addresses the points raised during the review process. Both reviewers raised several concerns, especially regarding the language and statistical analysis, that needed to be addressed. You need to effectively respond to these comments in your revised manuscript.

We look forward to receiving your revised manuscript.

Kind regards,

Yu Ru Kou, PhD

Academic Editor

PLOS ONE

Journal Requirements:

Reviewers' comments:

Reviewer's Responses to Questions

**Comments to the Author**

1. Is the manuscript technically sound, and do the data support the conclusions?

Reviewer #1: Partly

Reviewer #2: Yes

2. Has the statistical analysis been performed appropriately and rigorously? 

Reviewer #1: Yes

Reviewer #2: Yes

3. Have the authors made all data underlying the findings in their manuscript fully available?

Reviewer #1: Yes

Reviewer #2: Yes

4. Is the manuscript presented in an intelligible fashion and written in standard English?

Reviewer #1: Yes

Reviewer #2: Yes

5. Review Comments to the Author

Reviewer #1: Dear author,

1. In the Introduction part of manuscript: from line 42 to 46. "If patients receive PMV, they are usually excluded as candidates for extra-corporeal membranous oxygenation (ECMO), and with limited resources during a pandemic, it may be considered withdrawal of treatment [3]. The rapid surge of medical needs depletes ventilators and ICU beds, making the use of anesthetic machines instead of ventilators compulsory."

These two sentences are very poorly worded and I suggest they be changed to a more clear form. English should be improved in entire Manuscript.

2. In the Results section in Table 1. there is stated that the group with PMV had 4 patients with Ventilatory associated pneumonia versus zero patients in non-PMV group. That is a big difference, since pneumonia can affect the LUS score significantly and this study has a small sample size. I would recommend you mention this confounding factor in the Results section and discuss it in limitations part of your Discussion. Please specify at which days since admission to ICU these patients had VAP and the possibility of this affecting your research results and how did you approach this in statistical analysis.

3. In the limitations part of your Discussion you state :"Finally, a daily comparison between the LUS score and chest CT was not performed because we had extremely limited CT imaging data (almost only available on admission)" but in Clinical data and Outcomes of Methods section you clearly state: "CT was performed at the time of admission, 1 week after, and when the physician in charge deemed it necessary." Why weren't LUS score and CT score compared on day-1 and day-7?

Reviewer #2: First of all, I'd like to congratule all the authors for the effort to carry out nice research during pandemic. It is certainly not easy to perform several lung ultrasounds in this scenario.

However the lung ultrasound score applied in the study was different from the usual score and potentially confusing in clinical practice. I'd suggest reviewing the statistical analysis for LUS separate from the pleural score and would also recommend focusing on their performance to predict successful extubation in the first place.

6. PLOS authors have the option to publish the peer review history of their article (what does this mean?). If published, this will include your full peer review and any attached files.

Reviewer #1: No

Reviewer #2: No

---

## [Author Response · Author response to Decision Letter 0]

10 Jun 2022

Responses to the Reviewers’ Comments

We thank you for critically analysing our manuscript and providing us invaluable comments and suggestions. Kindly find our point-by-point response to each of the comments below, which we believe has addressed all your concerns.

Kindly note that the page and line numbers in our responses below refer to those of our revised manuscript highlighted in red.

Reviewer #1: 

COMMENT 1-1: 

In the Introduction part of manuscript: from line 42 to 46. "If patients receive PMV, they are usually excluded as candidates for extra-corporeal membranous oxygenation (ECMO), and with limited resources during a pandemic, it may be considered withdrawal of treatment [3]. The rapid surge of medical needs depletes ventilators and ICU beds, making the use of anesthetic machines instead of ventilators compulsory."

These two sentences are very poorly worded and I suggest they be changed to a more clear form. English should be improved in entire Manuscript.

RESPONSE 1-1:

We appreciate your thoughtful comment and sincerely apologise for these sentences. 

We have revised the sentences in the introduction section of our revised manuscript.

We have also used the services of Editage editing firm to improve the entire Manuscript.

In the introduction section: Page 2, Lines 72-76

The longer the patient is on ventilatory management, the higher the risk of developing ventilator-induced lung injury, and the lung itself is more damaged [3]. Extracorporeal Membrane Oxygenation (ECMO) cannot save irreversibly damaged lungs and is therefore not recommended [4]. In addition, treatment discontinuation may be considered owing to limited resources during a pandemic [5].

COMMENT 1-2: 

In the Results section in Table 1. there is stated that the group with PMV had 4 patients with Ventilatory associated pneumonia versus zero patients in non-PMV group. That is a big difference, since pneumonia can affect the LUS score significantly and this study has a small sample size. I would recommend you mention this confounding factor in the Results section and discuss it in limitations part of your Discussion. Please specify at which days since admission to ICU these patients had VAP and the possibility of this affecting your research results and how did you approach this in statistical analysis.

RESPONSE 1-2: 

We thank the reviewer for pointing out this critical issue in our study. We have revised and added the following sentence to the Discussion section of our revised manuscript.

In the discussion section: Page 12, Lines 423– Page 13, Lines 452 

Regarding ventilator-associated pneumonia (VAP), only four patients were observed in the PMV group, while there was none in the non-PMV group, but this was not statistically different. In addition, the onset of VAP occurred after 1 week of admission, when our scorings were not performed.

COMMENT 1-3: 　 

In the limitations part of your Discussion you state :"Finally, a daily comparison between the LUS score and chest CT was not performed because we had extremely limited CT imaging data (almost only available on admission)" but in Clinical data and Outcomes of Methods section you clearly state: "CT was performed at the time of admission, 1 week after, and when the physician in charge deemed it necessary." Why weren't LUS score and CT score compared on day-1 and day-7?

RESPONSE 1-3:

We thank the reviewer for the insightful comment. We did not state correctly regarding the timing of CT. We sincerely apologise for th inaccurate description. 

We have revised the sentence in the Methods section and limitation part of our revised manuscript.

In the methods section: Page 4, Lines 181-183

CT was performed at admission and whenever the physician in charge deemed it necessary.

In the limitation part: Page 13, Lines 470- 475

Finally, we did not perform chest CT at day3 and 7 in all cases. Thus, to evaluate the progression of lung pathology, such as lung fibrosis, the daily comparisons between the LUS score and chest CT could not be performed in this study. Therefore, a longitudinal study that would continuously measure LUS scores and CT with progression of lung fibrosis is required in future studies.

Reviewer #2 

First of all, I'd like to congratule all the authors for the effort to carry out nice research during pandemic. It is certainly not easy to perform several lung ultrasounds in this scenario.

However the lung ultrasound score applied in the study was different from the usual score and potentially confusing in clinical practice. I'd suggest reviewing the statistical analysis for LUS separate from the pleural score and would also recommend focusing on their performance to predict successful extubation in the first place.

RESPONSE 2-1:

We appreciate the reviewer’s valuable suggestion. We agree with your comment and have added the assessment of LUS score without the pleural score to the Methods and Results, and Discussion sections, and in Tables 1, and 2 and Figures 2, 3 of our revised manuscript.

In the Methods section: Page 4, Lines 195– Page 5, Lines 219

In this study we used two types of LUS scores. One was the popular LUS score: score 0: A-lines or two or fewer well-spaced B-lines; score 1, three or more well-spaced B-lines; score 2, coalescent B-lines; score 3, tissue-like pattern, which were used to predict ARDS severity, progression, and lung reaeration in previous studies [16]. The sum of the scores in all 12 zones yielded a final score (ranging from 0 to 36). The other scoring system was modified-LUS (m-LUS) score, in which B-lines/consolidations were quantitatively scored as follows: score 0, well-spaced B-lines <3; score 1, well-spaced B-lines ≥3; score 2, multiple coalescent B-lines; and score 3, lung consolidation. The pleural line was quantitatively scored as follows: score 0, normal; score 1, irregular pleural line; and score 2, blurred pleural line, which were associated with COIVD-19 severity at admission [17]. The sum of both scores in all 12 zones yielded a final score with a range between 0 and 60. 

In the Result section: Page 6, Table 1

In the Result section: Page 8, Table 2

We added the LUS score, which is without the pleural score, in Table 1,2.

In the result section: Page 11.Lines 329–336

Analysis of the two types of serial LUS scores for each outcome.

Fig 3a shows the serial LUS and m-LUS scores on days 1, 3, and 7 for with the PMV groups. The m-LUS score on day 7 was significantly higher than that on day 1 (p<0.05), while the LUS score did not exhibit significant differences. Fig 3b shows the serial LUS and m-LUS scores on days 1, 3, and 7 with the successful extubation groups. The m-LUS score on day 7 was significantly lower than that of day 1. Though the LUS score on day3 and 7 was significantly lower than that of day 1, the LUS scores itself did not exhibit significant differences with or without successful extubation.

In the result section: Page 11, Figure 3

We revised Fig 3 and Figure title.

In the result section: Page 11, Figure 4

We revised Figure title.

In the discussion section: Page 11, Lines 349–351

This study showed that a higher m-LUS score on day 7 was a predictor for PMV, while a lower m-LUS score on day 7 was a predictor for successful extubation in patients with severe COVID-19.

In the discussion section: Page 12, Lines 418–423

This study showed no difference in water balance or cardiac function according to the outcome. Therefore, we believe that the worsening of serial LUS scores can be used to evaluate lung injury, such as fibrosis, and not wet lung. Moreover, we analysed two type serial LUS scores and showed that m-LUS score was more sensitive than LUS score for associating outcomes. We thought that m-LUS score, which assessed pleural line changes in detail, has more association with this study outcome.

---

## [Decision Letter · Decision Letter 1]

30 Jun 2022

Prolonged mechanical ventilation in patients with severe COVID-19 is associated with serial modified-lung ultrasound scores: A single-centre cohort study

PONE-D-22-08025R1

Dear Dr. Taniguchi,

We’re pleased to inform you that your manuscript has been judged scientifically suitable for publication and will be formally accepted for publication once it meets all outstanding technical requirements.

Kind regards,

Yu Ru Kou, PhD

Academic Editor

PLOS ONE

Additional Editor Comments (optional):

Reviewers' comments:

Reviewer's Responses to Questions

**Comments to the Author**

1. If the authors have adequately addressed your comments raised in a previous round of review and you feel that this manuscript is now acceptable for publication, you may indicate that here to bypass the “Comments to the Author” section, enter your conflict of interest statement in the “Confidential to Editor” section, and submit your "Accept" recommendation.

Reviewer #1: All comments have been addressed

2. Is the manuscript technically sound, and do the data support the conclusions?

Reviewer #1: Yes

3. Has the statistical analysis been performed appropriately and rigorously? 

Reviewer #1: Yes

4. Have the authors made all data underlying the findings in their manuscript fully available?

Reviewer #1: Yes

5. Is the manuscript presented in an intelligible fashion and written in standard English?

Reviewer #1: Yes

6. Review Comments to the Author

Reviewer #1: All comments have been appropriately adressed. They had some English issues which was corrected according to my suggestions.

7. PLOS authors have the option to publish the peer review history of their article (what does this mean?). If published, this will include your full peer review and any attached files.

Reviewer #1: No

---

## [Editor Report · Acceptance letter]

4 Jul 2022

PONE-D-22-08025R1 

Prolonged mechanical ventilation in patients with severe COVID-19 is associated with serial modified-lung ultrasound scores: A single-centre cohort study 

Dear Dr. Taniguchi:

I'm pleased to inform you that your manuscript has been deemed suitable for publication in PLOS ONE. Congratulations! Your manuscript is now with our production department. 

Kind regards, 

on behalf of

Dr. Yu Ru Kou 

Academic Editor

PLOS ONE